# Assessment of China's Offshore Wind Resources Based on the Integration of Multiple Satellite Data and Meteorological Data

**Qiaoying Guo [1,2,3], Ran Huang [4], Liwei Zhuang [5], Kangyu Zhang [1,2,3] and Jingfeng Huang [1,2,3,*]**

[1] Institute of Applied Remote Sensing and Information Technology, Zhejiang University, Hangzhou 310058, China; qiaoyingguo@zju.edu.cn (Q.G.); kangyuzhang@zju.edu.cn (K.Z.)

[2] Key Laboratory of Agricultural Remote Sensing and Information Systems, Zhejiang University, Hangzhou 310058, China

[3] State Key Laboratory of Satellite Ocean Environment Dynamics, Second Institute of Oceanography, State Oceanic Administration, Hangzhou 310012, China

[4] School of Automation, Hangzhou Dianzi University, Xiasha Higher Education Zone, Hangzhou 310018, China; ran_huang@hdu.edu.cn

[5] National Meteorological Center, Beijing 100081, China; zhuanglw@cma.gov.cn

[*] Correspondence: hjf@zju.edu.cn; Tel.: +86-571-8898-2830

**Abstract:** Wind resources assessment plays a significant role in site selection for the construction of offshore wind farms. Mean wind speeds (MWS), wind power densities (WPD), and Weibull parameters are the most important variables for wind resources assessment. These variables were estimated with the synergetic use of multiple satellite data (QuikSCAT + WindSAT + ASCAT) and meteorological data from coastal stations using spatial interpolation methods, including inverse distance weighting (IDW), ordinary kriging (OK), and ordinary co-kriging (OCK). The spatial variability of offshore wind energy resources over the China Sea is assessed at heights of 10 m and 100 m (hub height of wind turbine). Then, 8 buoy measurements were used to evaluate the accuracy of the offshore wind resources assessment. Our results show that combining multiple satellite data and coastal meteorological data improves the accuracy of wind resources assessment in the offshore areas and the OCK method show the best performance for accuracy in most cases. The statistical results comparing buoy-derived MWS and interpolated MWS show a root mean square error (RMSE) of 0.17 m/s and correlation coefficient (Corr.) of 0.987 at a height of 10 m. Statistics of the comparison between buoy-derived WPD and interpolated WPD by OCK show a RMSE of 23.38 W/m$^2$ at a height of 10 m. The results show that the highest wind resources are mainly found in the Taiwan Strait and offshore regions in Fujian province.

**Keywords:** offshore wind resources; spatial interpolation; QuikSCAT; WindSAT; ASCAT; meteorological data; China Sea

## 1. Introduction

China's economic growth and rapid urbanization has required substantial energy consumption. Traditional fossil fuels have a damaging impact on the environment. China needs to utilize renewable energy to ensure energy security and environmental sustainability. Wind energy has been one of the fastest growing renewable energy sources during the last decade. In China, the total installed wind energy capacity rose from 25.8 GW in 2009 to 211.4 GW in 2018. The newly installed wind energy capacity of China in 2018 was 23 GW, comprising nearly 45 percent of the world's new wind energy installations [1]. Due to the lower surface roughness of ocean compared with land, offshore

wind resources are higher than onshore wind resources. Coastal provinces in China have advanced economies and higher demand for energy than other regions, so taking full advantage of the offshore wind resources could save land resources and transportation costs [2]. The total installed capacity of global offshore wind power was 23.1 GW in 2018. Of this total, 4.6 GW of offshore installations in 2018 were in China, putting the country in third place globally, behind the United Kingdom (7.96 GW) and Germany (6.4 GW). China installed 1.8 GW of new offshore wind capacity in 2018, taking the world's lead for the first time, followed by the United Kingdom (1.3 GW) and Germany (0.97 GW) [1].

Accurate offshore wind information plays an important role in offshore wind energy assessment and planning [3]. In situ wind measurements used for offshore wind energy resources assessment [4–12] are usually limited and sparse, involving coastal stations, buoys, ships, masts, and oil platforms. Due to the development of microwave remote sensing, previous studies revealed that sea surface wind data derived from satellite data have been applied to offshore and ocean wind resources assessment, including sea surface wind data derived from synthetic aperture radars (SAR) [13–24], scatterometers [3,19,23–40], and radiometers [39–42]. The low temporal resolution (3–7 images each month) of SAR leads to less overlapping of samples [24]. Ocean wind fields retrieved from scatterometers and radiometers have higher temporal resolution (two observations per day). Previous researches pointed out that the accuracy of wind resources estimation could be improved with the synergetic use of multiple satellite data due to the increasing number of observations and different equatorial crossing times [24,39,43].

Spatial interpolation can be used to obtain onshore and offshore wind information [27,36,37,40,44–47]. Deterministic interpolation methods create surfaces from measured points and can either force the resulting surface to pass through the data values or not. Geostatistical interpolation methods quantify the spatial autocorrelation among measured points and account for the spatial configuration of the measured points around the prediction location [48]. The kriging and inverse distance weighting (IDW) methods have been used in the interpolation of wind information and kriging methods were widely used [47]. In this study, spatial interpolation methods included one deterministic interpolation method (IDW) and two geostatistical interpolation methods (ordinary kriging (OK) and ordinary co-kriging (OCK)).

Offshore wind information retrieved from satellite data in nearshore regions are not as accurate as those at open sea due to land contamination, and satellite wind products usually mask wind information near coastlines. Therefore, the accuracy of offshore wind resources estimation using satellite data might also be affected [3,23,31,42]. The purpose of this work is to estimate the accuracy of offshore wind resources assessment based on synergetic use of multiple satellite data and meteorological data from coastal stations using spatial interpolation methods. The spatial variability of offshore wind energy resources over the China Sea is assessed at heights of 10 m and 100 m (hub height of wind turbine).

## 2. Data

### 2.1. Satellite Data

This study relies on two types of satellite data from two scatterometers (QuikSCAT and ASCAT) and one radiometer (WindSAT), namely the Ku-band (13.4 GHz) SeaWinds scatterometer onboard QuikSCAT satellite, C-band (5.3 GHz) ASCAT scatterometer onboard Metop-A satellite, and WindSAT fully polarimetric radiometer onboard Coriolis satellite. Wind products used in this study are the daily gridded maps from Remote Sensing Systems [49]. They can provide sea surface wind information over oceans with a spatial resolution of $0.25° \times 0.25°$ at 10 m above sea level. There are at most two observations from a single satellite sensor per day. The local ascending node time of a satellite is maintained at about 18:00 (QuikSCAT and WindSAT) and 21:30 (ASCAT). The local descending node time of a satellite is maintained at about 06:00 (QuikSCAT and WindSAT) and 09:30 (ASCAT) [39,49]. We removed rain effects from the satellite datasets using the rain flags. Further details about the three satellite data can be found in previous studies [39].



According to previous studies, with the synergetic use of multiple satellite observations, the accuracy of wind resources estimation may improve due to the increasing number of observations [24,39,43]. Figure 1 shows the number of overlapping samples from combining QuikSCAT (1999–2009), WindSAT (2003–2017), and ASCAT (2007–2017) data for the China Sea. The number of samples from all of the satellite data is greater than 8000 in order to maintain the accuracy of wind resources estimation (shown in Figure 1). The number of satellite data from the offshore areas near the coastline is much lower than that from the open sea because of the limitations in retrieving the wind vectors from satellite observations near the coast. There are almost no satellite data in some offshore areas (such as Hangzhou Bay). The total number of satellite data is about 8000–14,000 from 0° N to 27° N, and is about 8000–15,000 among all satellite data at 27–32° N. The total number of satellite data is about 8000–17,000 at 32–41° N.

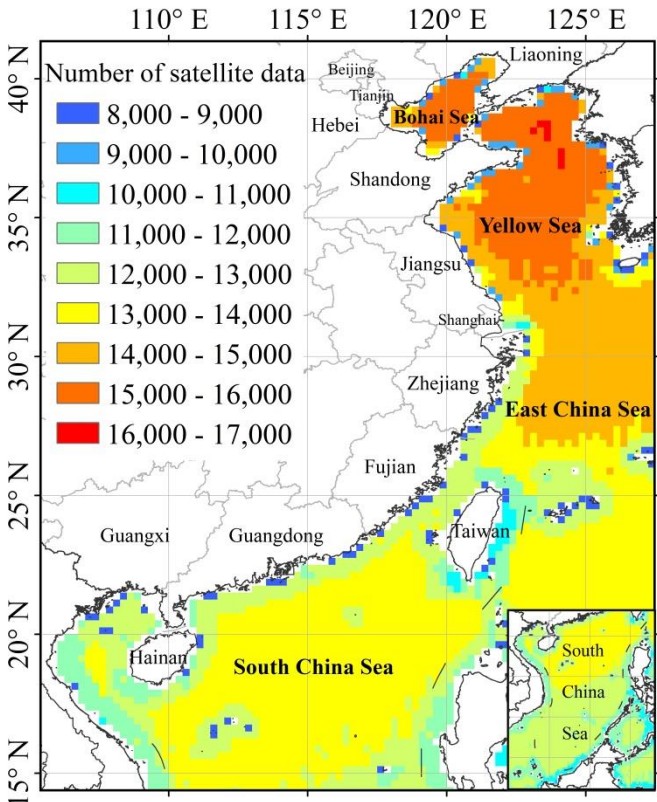

**Figure 1.** Number of overlapping samples combined from QuikSCAT (1999–2009), WindSAT (2003–2017), and ASCAT (2007–2017) data for the China Sea.

*2.2. Meteorological Data*

The China Meteorological Administration provides average hourly wind vector measurements, which are the 10 min average wind vectors measured at the top of every hour, recorded at 8 buoys over the China Sea and 480 meteorological stations within 10 km distance from the coastline along the coastal areas of China (shown in Figure 2). The 8 buoy measurements were selected as validation data and they provide one or two years of meteorological measurements at a height of 10 m above sea level [50]. All buoys are equipped with propeller anemometers. Table 1 summarizes the information of buoy measurements, including the number of buoy measurements. The number of buoy measurements range from 7586 to 17,241. The distances from buoys to coastline are mainly less than 60 km, except for buoy 59765, which is located up to 86 km from the coastline. The water depths of the buoys' positions range from 27 m to 55 m.

The 480 meteorological stations were selected as interpolation data and they provide one or two years of meteorological measurements at 10 m height for land located within 10 km distance from the coastline from 2016 to 2017. The elevations of 480 meteorological stations are less than 50 m, and there are no mountains along the coastal side of these selected stations. The number of measurements from 261 meteorological stations range from 7902 to 8779 over one year. The number of measurements from 219 meteorological stations range from 16,218 to 17,539 over two years.

Among the 480 selected meteorological stations, we chose the meteorological stations that were located on the coastline of China using the following rules: when the spatial distribution of meteorological stations was relatively dense (there are more than 2 stations within 25 km along the coastline), we chose the stations which are near the coastline (less than 1 km distance from the coastline). When the spatial distribution of meteorological stations was relatively sparse, we chose the station which is relatively closer to the coastline (less than 5 km distance from the coastline). The total number of meteorological stations located along the coastline of China is 270 (shown in Figure 2).

The wind profile method (in Section 3.1) was used to extrapolate wind speeds to 100 m height for interpolation of wind resources at the hub height of the wind turbines.

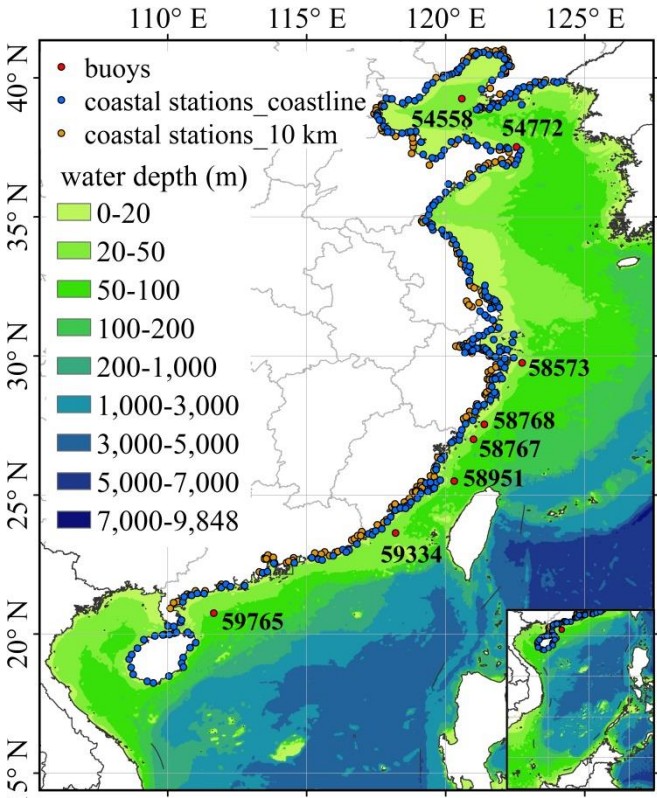

**Figure 2.** Geographical location of meteorological stations and buoys.

**Table 1.** Information for buoy measurements.

| Buoy | Water Depth (m) | Distance to Coastline (km) | Time of Datasets | Number of Measurements |
|---|---|---|---|---|
| 54558 | 33 | 59 | 01.2011–12.2011 | 8440 |
| 54772 | 33 | 12 | 01.2016–12.2016 | 8680 |
| 58573 | 27 | 35 | 01.2011–12.2012 | 16,231 |
| 58767 | 35 | 52 | 12.2015–11.2016 | 7586 |
| 58768 | 32 | 40 | 01.2012–12.2012 | 8482 |
| 58951 | 55 | 45 | 01.2017–12.2017 | 8489 |
| 59334 | 40 | 55 | 01.2016–12.2017 | 17,241 |
| 59765 | 48 | 86 | 01.2016–12.2017 | 16,396 |

### 2.3. Elevation and Bathymetric Data

Digital elevation model (DEM) data on land were obtained from the shuttle radar topography mission (SRTM) project, with a spatial resolution of 3 arc-second. The DEM data were resampled to $0.01° \times 0.01°$ by pixel averaging. The bathymetric data over the China Sea were obtained from the National Geophysical Data Center (NGDC), called the ETOPO1 1 arc-minute global relief model. The ETOPO1 data were also resampled to $0.01° \times 0.01°$ using the nearest neighbor method. The DEM was used on land and bathymetric data over the China Sea as an auxiliary variable for the OCK spatial interpolation method.

## 3. Methods

This study focuses on the spatial representation of offshore wind resources assessment (1999–2017). We synergetically used multiple satellite data (QuikSCAT + WindSAT + ASCAT) and meteorological data from coastal stations to obtain MWS, WPD, and Weibull parameters based on three spatial interpolation methods (IDW, OK, OCK), with a spatial resolution of $0.01° \times 0.01°$ at heights of 10 m and 100 m over the China Sea. In order to estimate the accuracy of offshore wind resources assessment, the MWS, WPD, and Weibull parameters calculated from the in situ measurements from 8 buoys at a height of 10 m were compared with those interpolated results derived from data from 480 meteorological stations within 10 km in the coastal areas of China, data from 270 meteorological stations along the coastline of China, and data from multiple satellites (QuikSCAT + WindSAT + ASCAT) and multiple satellites + 480/270 meteorological stations, respectively.

The root mean square error (RMSE), mean absolute error (MAE), bias, and correlation coefficient (Corr.) were used to compare interpolated MWS, WPD, and Weibull parameters with 8 buoy-derived MWS, WPD, and Weibull parameters at a height of 10 m.

### 3.1. Extrapolating Wind Speed to Hub Height

In order to extrapolate wind speeds to turbine hub height (100 m), the wind speeds can be calculated using a logarithmic profile method:

$$V(z) = \frac{u_*}{\kappa} \left[ \ln\left(\frac{z}{z_0}\right) - \psi_m \right] \tag{1}$$

$$z_0 = \alpha_c \frac{u_*^2}{g} \tag{2}$$

where $V$ is the wind speed at a height of $z$ (100 m), $u_*$ is the friction velocity, $\kappa$ is the von Karman's constant (~0.4), and $\psi_m$ is the factor of atmospheric stability correction, which is considered zero in neutral stability conditions [39,51]; further details about the wind speed extrapolation impacted by $\psi_m$ can be found in previous studies [39]. Here, $z_0$ is the roughness length, $\alpha_c$ is Charnock's parameter (~0.0144) [51], and $g$ is the gravitational acceleration of the Earth. When the wind speed at a single level is known, Equations (1) and (2) can be combined and solved iteratively to estimate the $u_*$ [51].

### 3.2. Wind Resources Assessment Method

The wind power density $E$ (W/m$^2$) is used to measure the theretical potential of wind resources at a particular place [52], and can be estimated using the following method:

$$E = \frac{1}{2n} \sum_{i=1}^{n} \rho V_i^3 \tag{3}$$

where $n$ is the number of wind speed observations, $\rho$ is the air density (~1.225 kg/m$^3$) [34,36,38,39,53], $\rho$ is the property of the wind measurement environment (air pressure, air temperature, and relative

humidity) [54], and the spatio-temporal variability of $\rho$ is ignored due to the lack of relevant data using standard sea-level air density [53,55]; $V_i$ is the wind speed at point *i*.

The Weibull distribution is the most commonly used statistical distribution for describing wind speed data at a fixed site [6]. There are several methods used to derive Weibull parameters [56]. In this study, the scale parameter C (m/s) and shape parameter k (dimensionless) are determined by using mean wind speed $\overline{V}$ (m/s) and standard deviation of wind speed $\sigma$ [26,34,56], as follows:

$$k = \left(\sigma/\overline{V}\right)^{-1.086} \tag{4}$$

$$C = \overline{V}/\Gamma(1 + \frac{1}{k}) \tag{5}$$

where $\Gamma$ is the Gamma function.

## 4. Results

### 4.1. Validation of Interpolated MWS and WPD

Table 2 reveals that interpolated MWS derived from data from multiple satellites + 270 meteorological stations using OCK method have the best accuracy in terms of RMSE (0.17 m/s), MAE (0.129 m/s), and Corr. (0.987). Interpolated MWS derived from meteorological station data underestimated the buoy-derived MWS in terms of negtive biases. Interpolated MWS derived from multiple satellite data overestimated the buoy-derived MWS in terms of small positive biases. Interpolated MWS derived from multiple satellites + 270/480 meteorological stations show lower errors than those from multiple satellite data or meteorological data only, in terms of lower RMSE and MAE and higher correlations. Interpolated MWS derived from the same dataset using OCK method show the best accuracy performance in most cases, followed by OK and IDW methods. The results of this comparison show that combining multiple satellite data and coastal meteorological data may improve the accuracy of interpolated MWS in the offshore areas.

**Table 2.** Statistics of the comparison between buoy-derived mean wind speeds (MWS) and interpolated MWS at a height of 10 m.

| Dataset for Interpolation | Interpolation Method | RMSE (m/s) | MAE (m/s) | Bias (m/s) | Corr. |
|---|---|---|---|---|---|
| 480 meteorological data | IDW | 2.456 | 2.347 | −2.347 | 0.686 |
| | OK | 2.078 | 1.919 | −1.919 | 0.608 |
| | OCK | 1.889 | 1.675 | −1.675 | 0.551 |
| 270 meteorological data | IDW | 2.121 | 1.924 | −1.924 | 0.600 |
| | OK | 1.749 | 1.593 | −1.593 | 0.692 |
| | OCK | 1.603 | 1.429 | −1.415 | 0.678 |
| Satellite data | IDW | 0.235 | 0.199 | 0.049 | 0.976 |
| | OK | 0.234 | 0.179 | 0.082 | 0.977 |
| | OCK | 0.230 | 0.178 | 0.081 | 0.977 |
| Satellite + 480 meteorological data | IDW | 0.214 | 0.177 | 0.054 | 0.980 |
| | OK | 0.188 | 0.153 | 0.002 | 0.988 |
| | OCK | 0.202 | 0.160 | −0.004 | 0.984 |
| Satellite + 270 meteorological data | IDW | 0.206 | 0.166 | 0.064 | 0.981 |
| | OK | 0.177 | 0.132 | 0.076 | 0.987 |
| | OCK | 0.170 | 0.129 | 0.065 | 0.987 |

Here, IDW = inverse distance weighting; OK = ordinary kriging; OCK = ordinary co-kriging; RMSE = root mean square error; MAE = mean absolute error; Corr. = correlation coefficient.

Table 3 indicates that interpolated WPD derived from multiple satellites + 270 meteorological stations using OCK method have the best accuracy in terms of RMSE (23.38 W/m$^2$) and MAE (16.88 W/m$^2$). Interpolated WPD underestimated the buoy-derived WPD in terms of negative biases. Interpolated WPD derived from multiple satellites + 270/480 meteorological stations show lower errors than those from multiple satellite data or meteorological data only in terms of lower RMSE and MAE. Interpolated WPD derived from the same dataset using OCK method show the best performance of accuracy in most cases, followed by OK and IDW methods. The result of this comparison shows that combining multiple satellite data and coastal meteorological data may improve the accuracy of interpolated WPD in the offshore areas.

**Table 3.** Statistics of the comparison between buoy-derived wind power densities (WPD) and interpolated WPD at a height of 10 m.

| Dataset for Interpolation | Interpolation Method | RMSE (W/m$^2$) | MAE (W/m$^2$) | Bias (W/m$^2$) | Corr. |
|---|---|---|---|---|---|
| 480 meteorological data | IDW | 266.93 | 239.37 | −239.37 | 0.689 |
| | OK | 240.19 | 208.97 | −208.97 | 0.705 |
| | OCK | 237.01 | 205.82 | −205.82 | 0.708 |
| 270 meteorological data | IDW | 245.99 | 217.76 | −217.76 | 0.710 |
| | OK | 227.32 | 197.56 | −196.45 | 0.730 |
| | OCK | 223.10 | 195.25 | −190.77 | 0.711 |
| Satellite data | IDW | 36.47 | 28.08 | −15.96 | 0.992 |
| | OK | 30.79 | 23.11 | −10.61 | 0.992 |
| | OCK | 30.63 | 23.00 | −11.10 | 0.992 |
| Satellite + 480 meteorological data | IDW | 25.33 | 19.12 | −13.09 | 0.995 |
| | OK | 24.98 | 20.11 | −14.98 | 0.996 |
| | OCK | 25.28 | 21.18 | −12.37 | 0.991 |
| Satellite + 270 meteorological data | IDW | 24.95 | 17.94 | −11.10 | 0.996 |
| | OK | 23.70 | 17.20 | −8.42 | 0.995 |
| | OCK | 23.38 | 16.88 | −8.45 | 0.995 |

## 4.2. Validation of Interpolated Weibull Parameters

Table 4 reveals that interpolated Weibull C derived from multiple satellite data + 270 meteorological stations using OK and OCK methods have the best accuracy in terms of RMSE (0.214 m/s) and Corr. (0.985). Interpolated Weibull C derived from meteorological data underestimated the buoy-derived Weibull C in terms of negtive biases. Interpolated Weibull C derived from multiple satellite data overestimated the buoy-derived Weibull C in terms of small positive biases. Interpolated Weibull C derived from multiple satellite data + 270/480 meteorological station data show lower errors than those from multiple satellite data or meteorological data only in terms of lower RMSE, MAE and higher correlations. Interpolated Weibull C derived from the same dataset using OCK method show the best performance of accuracy in most cases, followed by OK and IDW methods. The results of this comparison show that combining multiple satellite data and coastal meteorological data may improve the accuracy of interpolated Weibull C in the offshore areas.

Table 5 indicates that interpolated Weibull k derived from multiple satellite data + 480 meteorological stations using OCK method has the best accuracy in terms of RMSE (0.156), MAE (0.140), and Corr. (0.873). Interpolated Weibull k derived from meteorological data underestimated the buoy-derived Weibull k in terms of negtive biases. Interpolated Weibull k derived from multiple satellite data overestimated the buoy-derived Weibull k in terms of positive biases. Interpolated Weibull k derived from multiple satellite data + 270/480 meteorological station data shows lower errors than those from multiple satellite data or meteorological data only in terms of lower RMSE, MAE, biases, and higher correlations. Interpolated Weibull k derived from the same dataset using OCK method shows the best performance for accuracy in most cases, followed by OK and IDW method.

The result of this comparison shows that combining multiple satellite data and coastal meteorological data may improve the accuracy of interpolated Weibull k data in the offshore areas.

**Table 4.** Statistics of the comparison between buoy-derived Weibull scale parameter (C) data and interpolated Weibull C data at a height of 10 m.

| Dataset for Interpolation | Interpolation Method | RMSE (m/s) | MAE (m/s) | Bias (m/s) | Corr. |
|---|---|---|---|---|---|
| 480 meteorological data | IDW | 2.793 | 2.669 | −2.669 | 0.689 |
| | OK | 2.359 | 2.178 | −2.178 | 0.607 |
| | OCK | 2.146 | 1.901 | −1.901 | 0.549 |
| 270 meteorological data | IDW | 2.416 | 2.190 | −2.190 | 0.596 |
| | OK | 1.994 | 1.816 | −1.816 | 0.690 |
| | OCK | 1.824 | 1.629 | −1.610 | 0.677 |
| Satellite data | IDW | 0.286 | 0.241 | 0.062 | 0.971 |
| | OK | 0.289 | 0.221 | 0.101 | 0.972 |
| | OCK | 0.285 | 0.219 | 0.098 | 0.972 |
| Satellite + 480 meteorological data | IDW | 0.263 | 0.217 | 0.067 | 0.976 |
| | OK | 0.237 | 0.178 | 0.006 | 0.984 |
| | OCK | 0.240 | 0.180 | 0.004 | 0.983 |
| Satellite + 270 meteorological data | IDW | 0.256 | 0.205 | 0.079 | 0.977 |
| | OK | 0.214 | 0.160 | 0.082 | 0.985 |
| | OCK | 0.214 | 0.161 | 0.080 | 0.985 |

**Table 5.** Statistics of the comparison between buoy-derived Weibull shape parameter (k) data and interpolated Weibull k data at a height of 10 m.

| Dataset for Interpolation | Interpolation Method | RMSE | MAE | Bias | Corr. |
|---|---|---|---|---|---|
| 480 meteorological data | IDW | 0.335 | 0.292 | −0.253 | 0.375 |
| | OK | 0.322 | 0.279 | −0.223 | 0.211 |
| | OCK | 0.298 | 0.252 | −0.195 | 0.302 |
| 270 meteorological data | IDW | 0.325 | 0.281 | −0.238 | 0.367 |
| | OK | 0.323 | 0.283 | −0.223 | 0.201 |
| | OCK | 0.286 | 0.238 | −0.201 | 0.556 |
| Satellite data | IDW | 0.174 | 0.160 | 0.115 | 0.852 |
| | OK | 0.178 | 0.160 | 0.112 | 0.832 |
| | OCK | 0.181 | 0.166 | 0.122 | 0.843 |
| Satellite + 480 meteorological data | IDW | 0.163 | 0.149 | 0.104 | 0.866 |
| | OK | 0.157 | 0.142 | 0.098 | 0.873 |
| | OCK | 0.156 | 0.140 | 0.097 | 0.873 |
| Satellite + 270 meteorological data | IDW | 0.165 | 0.151 | 0.106 | 0.861 |
| | OK | 0.164 | 0.149 | 0.105 | 0.862 |
| | OCK | 0.163 | 0.148 | 0.104 | 0.861 |

*4.3. Spatial Variability of Interpolated Offshore Wind Resources over the China Sea*

The MWS, WPD, and Weibull C at heights of 10 m and 100 m were calculated from multiple satellite data during 1999–2017 over the China Sea and from data from 270 meteorological stations. The Weibull k at heights of 10 m and 100 m were calculated from multiple satellite data during 1999–2017 over China Sea and from data from 480 meteorological stations. OCK method was used to acquire the MWS, WPD, and Weibull C and k with a spatial resolution of 0.01° × 0.01° over the China Sea.

The geographic distribution of interpolated MWS, WPD, and Weibull parameters at 10 m height above sea level with a spatial resolution of $0.01° × 0.01°$ over the China Sea is shown in Figure 3. It can be observed that the MWS and WPD in most areas of the China Sea are higher than 5 m/s and 200 W/m$^2$, respectively. Figure 3a,b shows that the highest MWS and WPD are mainly found in the Taiwan Strait and offshore regions in Fujian province. This result is consistent with those of existing studies [3,52,57,58]. The MWS and WPD in most areas of the Taiwan Strait are higher than 8 m/s and 500 W/m$^2$, respectively. The offshore areas in the East China Sea and the South China Sea have higher MWS and WPD than those areas in the Bohai Sea and the Yellow Sea. The MWS and WPD are mainly 5–10 m/s and 200–950 W/m$^2$ within 50 m depth off the coast of Fujian province. The MWS and WPD within 50 m depth in the offshore areas of Guangdong province are mainly 4–8.5 m/s and 150–650 W/m$^2$, respectively. The MWS and WPD within 50 m depth in the offshore areas of Zhejiang province are mainly 4–8 m/s and 150–500 W/m$^2$, respectively. The MWS and WPD in the Bohai Sea and the Yellow Sea are 4–6.5 m/s and 150–300 W/m$^2$, and 4–7.5 m/s and 150–450 W/m$^2$, respectively.

The spatial variability of interpolated Weibull C at 10 m height above sea level over the China Sea is similar with that of MWS at 10 m height, and the absolute values of Weibull C are slightly higher than that of MWS at 10 m height. The Weibull C parameters in most regions of the China Sea are mainly 6–11 m/s. The Weibull C parameters in most regions of the Taiwan Strait are 9–11 m/s. The Weibull C parameters are mainly 5.5–11 m/s within 50 m depth off the coast of the Fujian province. The Weibull C parameters within 50 m depth in the offshore areas of Guangdong province are mainly 5–9.5 m/s. The Weibull C parameters within 50 m depth in the offshore areas of Zhejiang province are mainly 5–9 m/s. The values of Weibull C in the Bohai Sea and the Yellow Sea are mainly 5–7 m/s and 5–8.5 m/s, respectively.

As shown in the Figure 3d, the values of interpolated Weibull k over the China Sea at 10 m above sea level are mainly 1.6–2.8. The values of Weibull k in the Bohai Sea and the Yellow Sea are mainly 1.6–2 and 1.6–2.4, respectively. The values of Weibull k in the East China Sea and the South China Sea are mainly 1.8–2.8 and 1.8–2.6, respectively.

The geographic distribution of interpolated MWS, WPD, and Weibull parameters at 100 m above sea level with a spatial resolution of $0.01° × 0.01°$ over the China Sea is shown in Figure 4. The spatial variability of interpolated MWS, WPD, and Weibull C at 100 m height are similar with those at 10 m height, but the absolute values are higher at 100 m than at 10 m. It can be observed that the MWS and WPD in most regions of the China Sea are higher than 6 m/s and 300 W/m$^2$, respectively. The Taiwan Strait have the highest wind resources in terms of the MWS and WPD, which are mainly 9–12 m/s and 700–1900 W/m$^2$, respectively. The MWS and WPD are mainly 6–12 m/s and 400–1800 W/m$^2$ within 50 m depth off the coast of Fujian province. The offshore MWS and WPD within 50 m depth across Guangdong province are mainly 6–10.5 m/s and 300–1200 W/m$^2$, respectively, a result similar to that found by Chang et al. [23], based on ENVISAT ASAR and ASCAT data from weather research and forecasting (WRF) model simulations, and by Hasager et al. [42], who used SSM/I data and WRF. The offshore MWS and WPD within 50 m depth across the Zhejiang province are mainly 6–9.5 m/s and 300–900 W/m$^2$, respectively. The MWS and WPD in the Bohai Sea and the Yellow Sea are mainly 6–7.5 m/s and 300–600 W/m$^2$, and 6–9 m/s and 300–800 W/m$^2$, respectively. The MWS are similar and WPD are slightly higher than the results obtained by Li et al. [59] based on the COSMO-CLM regional climate model in the Bohai Sea and the Yellow Sea.

The spatial variability of interpolated Weibull C at 100 m above sea level over the China Sea is similar with that of MWS at 100 m height, and the absolute values of Weibull C are slightly higher than that of MWS at 100 m height. The Weibull C parameters in most regions of the China Sea are mainly 7–13.5 m/s. The Weibull C parameters in most regions of the Taiwan Strait are 10–13.5 m/s. The Weibull C parameters are mainly 7–13.5 m/s within 50 m depth off the coast of Fujian province. The Weibull C parameters within 50 m depth in the offshore areas of Guangdong province are mainly 7–12 m/s. The Weibull C parameters within 50 m depth in the offshore areas of Zhejiang province are

mainly 7–11 m/s. The values of Weibull C in the Bohai Sea and the Yellow Sea are mainly 7–9 m/s and 7–10 m/s, respectively.

As shown in Figure 4d, the values of interpolated Weibull k over the China Sea at 100 m above sea level are mainly 1.6–2.6. The values of Weibull k in the Bohai Sea and the Yellow Sea are mainly 1.6–2 and 1.6–2.2, respectively. The values of Weibull k in the East China Sea and the South China Sea are mainly 1.8–2.6 and 1.8–2.6, respectively.

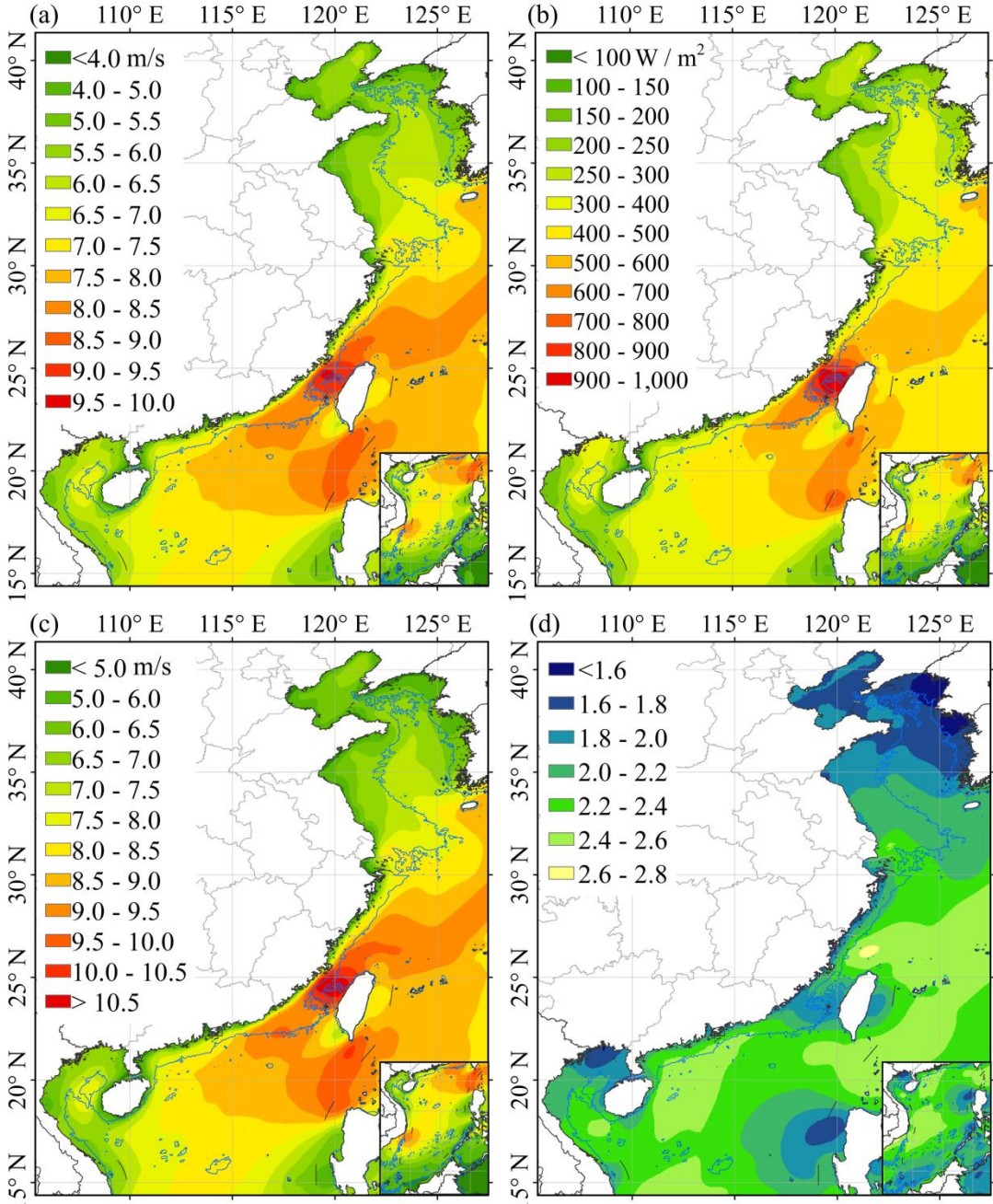

**Figure 3.** The geographic distribution of interpolated MWS, WPD, and Weibull parameters at 10 m above sea level using OCK method with a spatial resolution of 0.01° × 0.01° over the China Sea (the blue line indicates water depth at 50 m): (**a**) interpolated MWS, (**b**) interpolated WPD, (**c**) interpolated Weibull C, and (**d**) interpolated Weibull k.

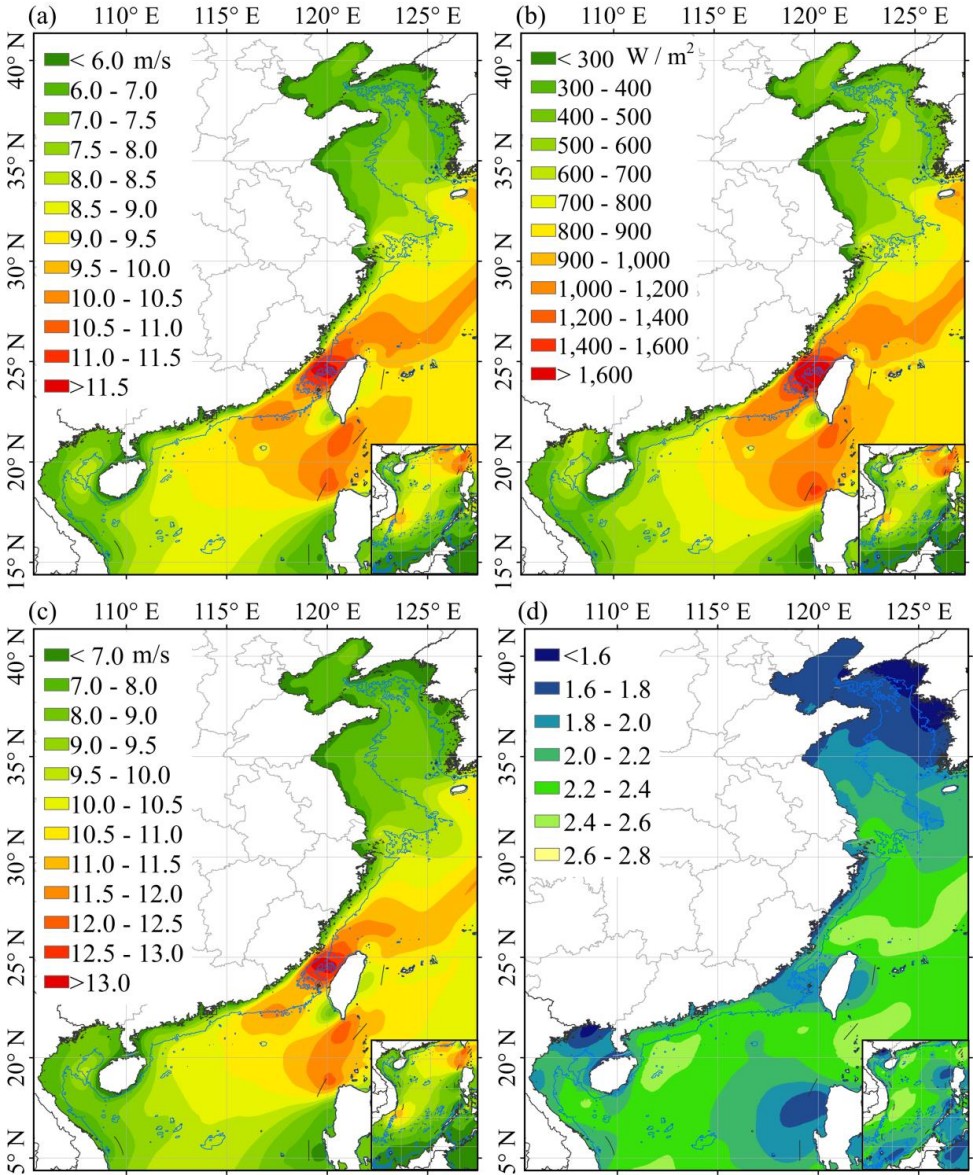

**Figure 4.** The geographic distribution of interpolated MWS, WPD, and Weibull parameters at 100 m above sea level using OCK method with a spatial resolution of 0.01° × 0.01° over the China Sea (the blue line indicates water depth at 50 m). (**a**) interpolated MWS, (**b**) interpolated WPD, (**c**) interpolated Weibull C, and (**d**) interpolated Weibull k.

## 5. Discussion

Previous studies pointed out that the mean wind speed on land is lower than offshore due to the higher surface roughness of land; in other words, MWS and WPD will be increased by increasing the offshore distance to the coastline, and there is a relatively steep wind gradient along the coastline [24,42,60]. Therefore, the bias of wind speeds along the coastline compared with wind speeds in offshore areas is larger than those in the open sea. In our study, satellite data dominates the wind resources in the open sea and meteorological data dominates those in coastal areas; interpolated MWS, WPD, and Weibull parameters derived from multiple satellite data show lower errors than those from meteorological data. Interpolated MWS, WPD, and Weibull parameters derived from multiple satellite data + 270/480 meteorological data demonstrate lower errors than those from multiple satellite data or meteorological data only. These results are consistent with the previous studies.

In this study, 270 meteorological masts (along the coastline of China) provide a better estimation of the MWS, WPD, and Weibull-C parameters than the use of 480 meteorological masts (within 10 km distance from the coastline). This result may be due to higher surface roughness of land than ocean, as the wind speeds on land decrease by increasing the land distance from the coastline. The conclusion could be made that the accuracy of wind resources assessment may also be improved by using less meteorological stations that are only located on the coastline of China. This will make the interpolation methods more operable and practical in consideration of the difficulty in acquiring the meteorological data from meteorological departments.

The spatial variability of interpolated MWS and WPD demonstrated that the offshore wind resources are abundant over the East and South China Sea, especially in the Taiwan Strait. This result is consistent with those of previous studies [3,52,57,58,61]. The spatial distribution of interpolated MWS and WPD in the offshore areas of the South China Sea at 10 m and 100 m height are similar to that in the study by Chang et al. [23] based on ENVISAT ASAR and ASCAT data from WRF simulations. The MWS are similar and WPD are slightly higher than those from Li et al. [59] based on the regional climate model COSMO-CLM in the Bohai Sea and the Yellow Sea at 100 m height. These minor differences may be impacted by the wind speed extrapolation method and simulation methods.

The effect of atmospheric stability is ignored in this study due to the lack of relevant data. However, atmospheric stability also affects the accuracy of wind speed extrapolation and wind resources assessment at the hub height of wind turbines. In future research, the application of atmospheric stability information is encouraged.

The spatio-temporal variability of air density is ignored due to the lack of relevant data using standard sea-level air density. Previous studies pointed out that the relevant deviations of air density (from −10% to 10% during the different seasons at a global scale) are mainly in the middle-high latitudes [55], and the relevant deviations in air density are smaller over ocean than on land, so the fixed air density is not completely accurate for WPD estimation over the China Sea. In future research, the information of air pressure, air temperature, and moisture are considered to evaluate WPD using meteorological data, mesoscale models, satellite data, or reanalysis [55].

In future studies, we can estimate the actual potential of wind energy over the China Sea by considering the types of reference wind turbines, wind farm wake effects, meteorological phenomena (including boundary layer height, atmospheric stability, and air density), environmental factors (including terrain effect near the coast, the effect of breaking waves), hard targets and human activity at sea (including low-level jets, shipping routes, fish farms, birds path, submarine cables, oil and gas platforms, and conservation areas) for choosing suitable regions for offshore wind farm construction .

## 6. Conclusions

Offshore wind data retrieved from satellite observations might be affected by land contamination along the coastline, and therefore the accuracy of offshore wind resources assessment using satellite data might also be affected. In this study, three spatial interpolation methods were applied to interpolate MWS, WPD, and Weibull parameters over the China Sea using multiple satellite date (QuikSCAT + WindSAT + ASCAT) and meteorological data from coastal stations. Then, 8 Chinese buoy measurements were used to evaluate the accuracy of offshore wind resources assessment. The results of buoy validation show that interpolated MWS, WPD, and Weibull parameters derived from multiple satellite data + 480/270 meteorological satellite data show lower errors than those from multiple satellite data or meteorological data only. Interpolated MWS, WPD, and Weibull parameters derived from the same dataset using the OCK method show the best performance for accuracy in most cases, followed by OK and IDW methods. The results of these comparisons show that combining multiple satellite data and coastal meteorological data may improve the accuracy of offshore wind resources assessment compared to using satellite data or meteorological data only.

The spatial distribution of MWS, WPD, and Weibull parameters over the China Sea is assessed at heights of 10 m and 100 m, interpolated by multiple satellite data and meteorological data. The results show that the most wind resources are mainly found in the Taiwan Strait and offshore regions of Fujian province.

**Author Contributions:** J.H. conceived the original idea of the study, and designed, organized and supervised the entire investigation. Q.G. collected, processed, and analyzed the data, and wrote the article. R.H., L.Z., and K.Z. assisted in data processing and analysis.

**Funding:** This research was funded by the China Special Fund for Industrial and Scientific Research in the Public Interest (Meteorology), grant number GYHY201306050.

**Acknowledgments:** The authors would like to thank the Remote Sensing Systems for providing QuikSCAT, WindSAT, and ASCAT data, and the China Meteorological Administration for the provision of meteorological data.

**Conflicts of Interest:** The authors declare no conflict of interest.

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
