# Peer review of "Assessment of China’s Offshore Wind Resources Based on the Integration of Multiple Satellite Data and Meteorological Data"

_remotesensing, doi:10.3390/rs11222680_

Round 1
Reviewer 1 Report
Chapter's 2 title (Date) might be incorrect (perhaps: Input Data?)
Some additional information about the boys should be given, ie: how do they measure wind data (kind of wind sensors, height of measurement). And the most critical: if buoys measurement height was not 10m, how the extrapolation was performed from 1m to 10m? again with the log-law?
There is no information about the time step of the meteorological data. Is it 10min, 1h, 3h. Similarly, the time step of the satellite data is missing.
Author Response
Dear Reviewer: We are truly grateful for your critical comments and thoughtful suggestions on our manuscript. Based on these comments and suggestions, we have made careful modifications on the manuscript. We hope the new manuscript will meet this journal’s standard. Below, you will find our point-by-point responses to the reviewers’ comments/questions. Responses to the reviewer #1’s comments: We express our profound thanks and appreciation for your comments and suggestions. The following is the point-by-point response to your comments: 1. Some additional information about the buoys should be given, ie: how do they measure wind data (kind of wind sensors, height of measurement). And the most critical: if buoys measurement height was not 10m, how the extrapolation was performed from 1m to 10m? Again with the log-law? Response: Accepted. The 8 Chinese buoys are equipped with propeller anemometers and measure wind vectors at the height of 10 m above sea level according to the information from China Meteorological Administration and references [50]. Please see page 3 lines 112-114 in the revised manuscript. 2. There is no information about the time step of the meteorological data. Is it 10min, 1h, 3h. Similarly, the time step of the satellite data is missing. Response: Accepted. There are at most two observations from a single satellite sensor per day. The local ascending node time of satellite is maintained at about 6 pm (QuikSCAT and WindSAT) and 9:30 pm (ASCAT). The local descending node time of satellite is maintained at about 6 am (QuikSCAT and WindSAT) and 9:30 am (ASCAT) [39, 49]. All meteorological data provide hourly wind vectors, which are the 10-min average wind vectors measured preceding the top of every hour, so the time difference between satellite observations and meteorological measurements is always within 30 min. Please see page 2 lines 88-91 and page 3 lines 109-112 in the revised manuscript.
Reviewer 2 Report
The manuscript deals with a meticulously investigation of the wind and wind power density off-shore in China. It is a very well written manuscript and reads easily.
I have a few comments to the manuscript that I would like the authors to take into account.
Please discuss why 480 meteorological data (masts) provide a poorer estimation of the wind and wind power density than use of 270 meteorological masts. It seems contra-intuitive that fewer measurements provide a better fit. Provide more information on how the 270 masts were selected. In the analysis low low-level jets, that are known to frequently occur in coastal areas, are not a accounted for. I ask the authors to discuss low-level jets and other meteorological phenomae (e.g. boundary layer height, baroclinity, atmospheric stability, effect of breaking waves in the roughness etc.) that are unaccounted for, and in this way provide a discussion on the shortcomings of the method applied. Line 169. I note that a constant air density is used. The Chinese coastline stretches over several climate zones, an assumption of constant temperature is not likely appropriate. Please add this point to the discussion above on shortcoming of the method. Line 175, Eq.(4) is not exact but an approximation, please provide a reference, e. g. Boundary-Layer Meteorology (2016), 159, 329-348. Add units on tables 2, 3 and 4.Author Response
Dear Reviewer:
Thank you very much for your suggestions. We truly appreciate the comments which have substantially helped us to further improve on the contents and structure of our manuscript. The following is the point-by-point response to your comments/ questions.
Responses to the reviewer #2’s comments:
Please discuss why 480 meteorological data (masts) provide a poorer estimation of the wind and wind power density than use of 270 meteorological masts. It seems contra-intuitive that fewer measurements provide a better fit. Provide more information on how the 270 masts were selected.Response: Accepted.
Among the 480 selected meteorological stations, we chose the meteorological stations which were located at the coastline of China using the following rules: when the spatial distribution of meteorological stations is relatively dense (there are more than 2 stations within 25 km along the coastline), we choose the stations which are near the coastline (less than 1 km distance from the coastline). When the spatial distribution of meteorological stations is relatively sparse, we choose the station which is relatively closer to the coastline (less than 5 km distance from the coastline). The total number of meteorological stations located at the coastline of China is 270 (shown in Figure 2).
In this study, 270 meteorological masts (at the coastline of China) provide a better estimation of the MWS/WPD/Weibull-C than use of 480 meteorological masts (within 10 km distance from the coastline). This result may be due to higher surface roughness of land than ocean, the wind speeds on land would be decreased by increasing the land distance from the coastline. The conclusion could be made that the accuracy of wind resource assessment may also be improved using less meteorological stations only located at the coastline of China. It will make the interpolation methods more operability and practicality in consideration of the difficulty in acquiring the meteorological data from meteorological department.
Please see page 4 lines 125-132 and page 12 lines 340-347 in the revised manuscript.
In the analysis low low-level jets, that are known to frequently occur in coastal areas, are not accounted for. I ask the authors to discuss low-level jets and other meteorological phenome (e.g. boundary layer height, baroclinity, atmospheric stability, effect of breaking waves in the roughness etc.) that are unaccounted for, and in this way provide a discussion on the shortcomings of the method applied.Response: Accepted.
In the future study, we can estimate the actual potential of wind energy over China Sea by considering the types of reference wind turbines, wind farm wake effects, meteorological phenomenon (including boundary layer height, atmospheric stability and air density, etc.), environment factors (including terrain effect near the coast, the effect of breaking waves in the roughness, etc.), hard targets and human activity at sea (including low-level jets, shipping routes, fish farms, birds path, submarine cables, oil and gas platforms and conservation areas, etc.) for choosing the suitable regions of offshore wind farm construction.
Please see page 12 lines 368-375 in the revised manuscript.
Line 169. I note that a constant air density is used. The Chinese coastline stretches over several climate zones, an assumption of constant temperature is not likely appropriate. Please add this point to the discussion above on shortcoming of the method.Response: Accepted.
The spatio-temporal variability of air density is ignored due to the lack of relevant data using standard sea-level air density [53, 55]. Previous studies pointed out that the relevant deviations of air density (about -10%~10% during the different seasons at a global scale) are mainly in the middle-high latitudes [55], and the relevant deviations in air density are smaller over ocean than those on land, so the fixed air density is not completely accurate for WPD estimation over China Sea. In future research, the information of air pressure, air temperature and moisture are considered to evaluate WPD using meteorological data, mesoscale models, satellite data or reanalysis [55].
Please see page 5 lines 178-181 and page 12 lines 361-367 in the revised manuscript.
Line 175, Eq.(4) is not exact but an approximation, please provide a reference, e. g. Boundary-Layer Meteorology (2016), 159, 329-348. Add units on tables 2, 3 and 4.Response: Accepted.
There are several methods to derive Weibull parameters [56]. In this study, the scale parameter C (m/s) and shape parameter k (dimensionless) are determined by using mean wind speed (m/s) and standard deviation of wind speeds [26, 34, 56].
Please see page 6 lines 182-185 and tables 2, 3 and 4 in the revised manuscript.
